# Glyoxal Fixation Is Optimal for Immunostaining of Brain Vessels, Pericytes and Blood-Brain Barrier Proteins

**DOI:** 10.3390/ijms23147776

**Published:** 2022-07-14

**Authors:** Sithara Thomas, Jayanarayanan Sadanandan, Spiros L. Blackburn, Devin W. McBride, Ari Dienel, Sungha Hong, Hussein A. Zeineddine, Peeyush Kumar Thankamani

**Affiliations:** The Vivian L. Smith Department of Neurosurgery, University of Texas Health Science Center, 6431 Fannin St. MSB 7.147, Houston, TX 77030, USA; sithara.thomas@uth.tmc.edu (S.T.); jayanarayanan.sadanandan@uth.tmc.edu (J.S.); spiros.blackburn@uth.tmc.edu (S.L.B.); devin.w.mcbride@uth.tmc.edu (D.W.M.); ari.c.dienel@uth.tmc.edu (A.D.); sungha.hong@uth.tmc.edu (S.H.); hussein.a.zeineddine@uth.tmc.edu (H.A.Z.)

**Keywords:** glyoxal, paraformaldehyde, vascular staining, blood-brain barrier, tight junction, pericytes

## Abstract

Brain vascular staining is very important for understanding cerebrovascular pathologies. 4% paraformaldehyde is considered the gold standard fixation technique for immunohistochemistry and it revolutionized the examination of proteins in fixed tissues. However, this fixation technique produces inconsistent immunohistochemical staining results due to antigen masking. Here, we test a new fixation protocol using 3% glyoxal and demonstrate that this method improves the staining of the brain vasculature, pericytes, and tight junction proteins compared to 4% paraformaldehyde. Use of this new fixation technique will provide more detailed information about vascular protein expressions, their distributions, and colocalizations with other proteins at the molecular level in the brain vasculature.

## 1. Introduction

Exploring proteins in the cells is fundamental to understanding tissue and organ function. Microscopic imaging of these biomolecules provides information about their distribution, expression, function, and colocalization with other proteins. The standard method is based on the biochemistry of antibody labeling or fluorescent protein techniques. Innovative fluorescence-based technologies have been recently developed with the help of super-resolution microscopy to visualize proteins at the nano-molecular scale [1,2,3]. However, after the current gold standard paraformaldehyde (PFA) fixation, immunohistochemical (IHC) staining of these cellular proteins using antibody or fluorophore binding remains problematic [4].

The general objective of fixation is to preserve essential chemical and physical characteristics of tissue or cells by disabling proteolytic enzymes, inhibiting the growth and activity of microbes and altering characteristics of the cells or tissues at the molecular level to improve their mechanical strength (as well as their stability) [5,6,7]. For immunostaining, an ideal fixation protocol should not reduce the accessibility or affinity of antigens for antibodies or alter tissue or cell structure [8].

Cerebral vasculature has a distinct anatomy compared to peripheral vessels and is heterogeneous. The structural and molecular analyses of the cerebral vasculature are key to delivering essential information on cerebral circulation, blood-brain barrier (BBB), and injury. The technical limitations in staining vascular components may slow down the progress of cerebrovascular research. In brain vascular staining, PFA ensures protein immobilization and ultrastructure preservation but interferes with epitope recognition and penetration of antibodies, subsequently resulting in poor or inconsistent staining of proteins in the brain vessels [9,10,11]. It remains critical to overcome this limitation and an improved fixation method is required for generating consistent results on routine vascular stainings such as structural exposition, protein distribution, and expression.

Prior publications have recommended glyoxal as an alternative to PFA and report that it can act faster than PFA, cross-links proteins more effectively and improves the preservation of cellular morphology [12]. However, glyoxal fixation has not been tested for its efficiency in brain vascular staining. In this manuscript, we investigated glyoxal fixation as an alternative to PFA to determine whether glyoxal is a dependable fixative for staining structural brain vascular networks, associated pericytes, and BBB/tight junction proteins.

## 2. Materials and Methods

### 2.1. Experimental Animals

C57BL/6 mice (weighing 25–30 g and age of 8–12 weeks) of both sexes, obtained from Jackson Laboratories, Farmington, CT, USA, were used for this study. All mice were fed with a standard laboratory diet and water, maintained under standard laboratory conditions (temperature: 25 ± 2 °C, humidity: 60 ± 5%, 12 h dark/light cycle) with free access to a standard pellet diet and water *ad libitum*. Animal procedures were carried out under the oversight of the Animal Care and Use Committee of the University of Texas Health Science Center at Houston (protocol # AWC-19-0120) and in strict compliance with National Institutes of Health guidelines.

### 2.2. Materials Used

PFA (P6148) was purchased from Sigma-Aldrich (MilliporeSigma, Burlington, MA, USA). Phosphate-buffered saline (PBS) (CA008-050) was obtained from Gen DEPOT, Katy, TX, USA. Permount mounting medium (SP15-500), hydrochloric acid (A481212), and sodium hydroxide (S320-500) were purchased from Fisher Scientific, Waltham, MA, USA. Glyoxal solution A (16525 A) and glyoxal solution B (16525 B) were bought from Electron Microscopy Sciences, Hatfield, PA, USA. Gills hematoxylin (24243-500) was procured from Polysciences, Warrington, PA, USA, and eosin Y (H-3502) was procured from Vector Laboratories, Newark, CA, USA.

### 2.3. Preparation of PFA

4% PFA was prepared in PBS (pH 7.4) with stirring and heating to approximately 60 °C in a ventilated hood, then cooled and filtered. The pH was adjusted to 6.9 using 1 N HCl and stored at 4 °C for one month.

### 2.4. Preparation of Glyoxal

4 mL of 3% glyoxal was prepared by mixing 3.6 mL of solution A (16525 A, Electron Microscopy Sciences) with 313 µL of solution B (16525 B, Electron Microscopy Sciences).

### 2.5. Fixation and Sectioning

Mice were kept under general anesthesia (isoflurane: 3% induction and 1% maintenance with 100% O2 as a carrier) and perfused (intracardiac) first with 5 mL PBS and then with 5 mL of 4% PFA. After perfusion, whole-brain samples were collected and stored in 4% PFA at 4 °C overnight and then transferred into PBS at 4 °C for long-term storage.

For glyoxal fixation, after PBS perfusion the whole brain was collected, washed with PBS and kept in 3% glyoxal overnight at room temperature with rotation. For long-term storage, brain samples in glyoxal were stored at 4 °C.

Extracted brains were then sectioned into 35-μm thick free-floating coronal slices using a semiautomatic vibratome (LEICA VT 1000S, Leica Biosystems, Deer Park, IL, USA). Collected PFA sections were stored at 4 °C in PBS till staining was carried out. Glyoxal sections were either used for staining or stored in glyoxal at 4 °C for long-term use.

### 2.6. H & E Staining

Matching sections were taken from both PFA and glyoxal fixed brain samples into glass slides and sections were completely covered with hematoxylin for three minutes followed by rinsing in two changes in distilled water, each for 15 s, dipped in 100% ethanol for 10 s, drained off the excess ethanol and then eosin Y solution was added to the slides and incubated for 30 s. Then, sections were dehydrated using ascending grades of ethanol from 70% to 100%, cleared in xylene and mounted with permount mounting medium (SP15-100 Fisher Scientific, USA).

### 2.7. Immunostaining

Brain samples were taken from both glyoxal and PFA fixed sections and observed under the bright field microscope (ZEISS Discovery. 12 steREO, ZEISS Microscopy, Dublin, CA, USA) to ensure tissue morphology is retained through both fixation techniques. For IHC analysis, slices were washed thrice in PBS + 0.1% TritonX (PBST) and blocked for 1 h with 1% BSA. Each slice was then incubated overnight at 4 °C with gentle shaking in PBST with primary antibody. The following antibodies were used: laminin (1:200, Anti-Laminin α-2 Antibody (4H8-2): sc-59854; Santa Cruz Biotechnology, Santa Cruz, CA, USA), isolectin B4 (1:200, DL-1207 Vector Laboratories, California, USA), claudin 5 (1:200, 4C3C2; Invitrogen, Waltham, MA, USA) VE-cadherin (1:200, 36-1900; Invitrogen, MA, USA), aquaporin-4 (1:400, A2887; ABclonal, Woburn, MA, USA) occludin (1:250, 00241; BiCell Scientific, Maryland Heights, MO, USA) zonula occludens-1 (ZO1) (1:250, 00236; BiCell Scientific, MO, USA), Neural/glial antigen 2 (NG2) (1:200, ab5320; Abcam, Fremont, CA, USA) and platelet-derived growth factor receptor beta (PDGFRB) (1:200, 31695; Cell Signaling Technologies, Danvers, MA, USA).

The following day, the brain slices were washed thrice with PBST. Sections stained with conjugated antibodies (laminin and isolectin B4) were mounted on glass slides using DAPI Fluoromount-G (SouthernBiotech Birmingham, AL, USA) and imaged using fluorescent microscope (Leica Thunder imager). Non-conjugated antibodies (claudin5, VE-cadherin, aquaporin-4, occluding, ZO-1, NG2 and PDGFRB) were then probed with a biotin-conjugated secondary antibodies from Vector Laboratories in PBST with 1% serum (1:200) for 1 h. The following secondary antibodies were used: Goat Anti-Rabbit IgG Antibody (H + L), Biotinylated (BA-1000-1.5), Goat Anti-Mouse IgG Antibody (H + L) and Biotinylated (BA-9200-1.5). Slices were then washed thrice in PBS, followed by 1 h incubation with Streptavidin, DyLight^®^ 488/594 (SA-5488-1/SA-5549-1), washed with PBS, mounted and imaged using fluorescent microscope (Leica Thunder imager).

### 2.8. Quantitative Analysis

Quantification of vascular staining from three different regions of interest in the glyoxal fixed brain microscopic images and/or 4% PFA fixed brain microscopic images was carried out using AngioTool [13]. Results were calculated as vessel percentage area (% vessel detected/total area of the microscopic image) and expressed as mean ± SD (standard deviation). The difference between the groups was assessed by GraphPad Prism 6 using an unpaired t-test and significance was accepted at *p* ≤ 0.05.

## 3. Results

### 3.1. Brain Architecture Was Preserved Both in PFA and Glyoxal Fixation

To evaluate and compare the preservation of brain anatomy across both PFA and glyoxal fixation conditions, we used H & E staining. Both PFA (overnight at 4 °C) and glyoxal (overnight at room temperature) were able to stain efficiently by H & E staining. As evident in Figure 1, in whole coronal brain sections, brain structures, and morphology were well-preserved with both fixatives.

### 3.2. Glyoxal Fixation Is Optimal for Obtaining Consistent Vascular Staining in the Brain

Next, we compared the 3 widely used vessel markers: Isolectin B4, CD31, and laminin for endothelial or vascular staining in PFA or glyoxal fixed free-floating brain sections. As shown in Figure 2A, PFA fixed brain shows positive only IB4 immunostaining while CD31 and laminin didn’t show any positive signal. Conversely, immunohistochemistry of these three-vessel markers in glyoxal fixed brain sections showed consistent staining of endothelial cells/vessels (Figure 2B).

### 3.3. Glyoxal Fixation Is Better for Higher Quality Blood-Brain Barrier/Tight Junction Protein Immunostaining

Then, we tested the efficacy of glyoxal fixation in staining the BBB proteins including tight junctions (TJs), transporter and adherens junction. We tested the effectiveness of PFA and glyoxal fixation in visualizing three significant TJ proteins, ZO-1, occludin and claudin-5 [14]. None of these three antibodies showed specific binding in the PFA fixed brain sections (Appendix A). However, for the glyoxal-fixed brain sections, continuous filaments of TJ proteins, ZO-1, occludin and claudin-5 are more easily detected (Figure 3). We also tested the immunostaining of adherens junction protein VE-cadherin [15]. In the glyoxal fixed brain VE-cadherin staining specifically detects and localizes with the vascular staining. In PFA fixed samples VE-cadherin staining fails to consistently detect the antigen across the vasculature. Further, we compared the immunostaining for a BBB transporter aquaporin-4 [16] and found strong vascular-specific staining for both PFA and glyoxal-fixed brain samples (Figure 3). We did not observe any difference with or without the use of blocking buffer indicating there is very little non-specific binding during this staining (results not shown).

### 3.4. Efficacy of PFA or Glyoxal Fixation in Pericyte Staining

Having shown that glyoxal is a more effective fixative than PFA for vascular protein staining, we then investigated glyoxal’s efficiency in staining pericytes. We used two common pericyte markers, NG2 proteoglycan and PDGFRB [17], for the immunohistochemical staining for pericytes. Laminin was used as a vessel marker. The PFA fixed brain didn’t show any staining for pericyte markers, NG2 and PDGFRB (Appendix A), while in glyoxal fixed brain shows uniform staining for pericyte markers along the vessels. We observed that NG2 immunohistochemistry showed flat staining around the vessels. Consistent with its receptor nature, PDGFRB staining was observed as more specific, spotty, and sporadic (Figure 4).

### 3.5. Quantitative Analysis of Vasculature Staining Glyoxal vs. 4% PFA

Next, we attempted to quantify the vascular staining in glyoxal and 4% PFA fixed brains. First, we quantified and compared the vascular area for IB4, AQP4 and VE-cadherin, which showed positive staining for both 4% PFA and 3% glyoxal fixed brains (Figure 5A). No statistical significance was observed in IB4 and AQP4 staining between PFA and glyoxal fixed brain sections indicating these staining works in both the fixatives. Adherens junction protein VE-cadherin showed a significant increase in the percentage of vessels stained in glyoxal compared to PFA (Figure 5A). Further, we quantified the vessel percentage area for vascular/pericyte proteins including CD31, laminin, ZO-1, claudin-5 and occludin in glyoxal fixed brain sections. In glyoxal fixed brain sections, claudin-5 showed significantly low vessel percentage area compared to laminin and occludin showed a significant decrease in vessel percentage area compared to laminin and CD31 (Figure 5B).

## 4. Discussion

It is important to maintain the macroscopic architecture and microenvironment of the brain samples by following the appropriate fixation technique. In the current study, we compared PFA (4%) with glyoxal (3%) fixation in maintaining brain structure in situ. Previous studies by Bussolati et al., 2017 and Richter et al., 2018 [18,19] have reported that glyoxal has rapid penetration properties. However, we found that an overnight incubation is required for fixing the deep brain structures in glyoxal.

A few recent studies [19,20,21,22,23] have replaced PFA with glyoxal (3%), in tissue fixation due to the specific advantages associated with glyoxal over PFA. Advantages include its less toxic nature [24], faster reaction rates and selective control over cross-linking so that it can retain immunoreactivity characteristics and can skip the antigen retrieval treatments [18,25]. Here we investigated glyoxal for better brain fixation, staining of multi-proteins in the brain vasculature, and pericyte staining.

To visualize blood vessels, immunohistochemical techniques targeting endothelial markers such as cluster of differentiation (CD) 31 [26,27], basement membrane markers including laminin [28] and lectins that bind to the luminal part of the capillary endothelium through interactions with sugar residues [29] have been widely used. Prior reports [30,31,32] and from our lab experience, immunohistochemical vascular staining methods with 4% PFA fixed sections as well as a lower PFA (2%) concentration (results not shown) did not provide consistent results (Figure 2). Also, from our experiments and as reported before, even IB4 staining is not consistent in PFA fixed sections [33]. Intriguingly, glyoxal fixed brain shows staining for all of the three-vessel markers (Figure 2). Thus, glyoxal fixation of the brain can be recommended as an ideal approach for learning any structural brain vascular abnormalities.

An important and unique characteristic of cerebral vessels is BBB. BBB restricts the movement of molecules between the blood and brain thereby maintaining cerebral homeostasis and proper neuronal function. BBB is composed of brain endothelial cells, microglia, pericytes and a basement membrane (comprised of type IV collagen, laminin and fibronectin), surrounded by astrocytes end-feet ensheathing [34]. Leaky BBB is a serious concern in various CNS disorders such as stroke, brain tumors and neurodegenerative disorders [35]. A high-resolution reliable staining method is warranted for detailed knowledge of BBB proteins, their interactions and modifications as well as their distribution that maintains BBB integrity during normal physiology or disease. Endothelial cells contribute to the core of the BBB [36,37,38,39]. The barrier properties of these brain endothelial cells notably depend on tight junctions (TJ) between adjacent cells: TJs are dynamic structures consisting of several transmembranes and membrane-associated cytoplasmic proteins [40]. Moreover, several studies have reported leaky BBB is associated with the loss of integrity in TJs [41]. Our analysis suggests that the immunostainings performed after glyoxal fixation can more readily allow the identification of TJ proteins (Figure 3). Moreover, 40× magnification imaging revealed a continuous strand of this tight junction along the vessels. This demonstrates glyoxal can preserve the BBB identity after fixation. This is important as it aids in determining the TJ protein damage after a brain injury [42].

A possible explanation for the lack of visualization of the TJ and adherens junction proteins after PFA fixation may be the high crosslinking and shrinking that mask the antigens from binding with antibodies. Interestingly, AQP4 staining was visualized in both PFA and glyoxal consistently (Figure 3). This can be related to the location of this protein, AQP4 is a water channel protein and is expressed in brain perivascular astrocyte processes [43].

Further, we examined glyoxal fixation for perivascular staining. Pericytes were found embedded within the basement membrane both on straight sections and at branch points of capillaries, with projections extending from the soma to wrap around the underlying vessel [44]. In recent times, various vascular functions of pericytes have been recognized including regulation of cerebral blood flow, maintenance of BBB and control of vascular development and angiogenesis [45]. In the glyoxal fixed brain, both the pericyte markers, NG2 and PDGFRB showed positive signals and were specifically localized around the vessels (Figure 4). Conversely, both the staining was negative in PFA fixed sections. 

Finally, the quantification of vascular staining revealed no difference in IB4 and AQP4 between PFA and glyoxal fixed brains indicating both the fixatives work well for these markers. However, VE-cadherin staining showed a significant decrease in detection capacity in PFA fixed brain sections compared to glyoxal fixed brain sections. This difference in staining can be attributed to their nature of expression and antigen masking due to high crosslinking observed in PFA fixed brain [46]. To overcome antigen masking from the high crosslinking during PFA fixation, different antigen retrieval methods can be used, including high-temperature treatment in citrate buffer and enzymatic treatment with pepsin [33,46,47]. However, this adds time and cost to the procedure and this could also result in a non-uniform or inconsistent staining pattern due to differences in antigen exposure to retrieval methods.

In summary the above mentioned experiments highlights the advantages of glyoxal fixation over 4% PFA in staining the vascular proteins. Table 1 summarize the pros and cons of using 4% PFA and glyoxal.

## 5. Conclusions

In summary, the above evidence supports that despite some limitations, glyoxal is a better alternative to PFA as it is more efficient, specific, easy to fix, and highly valuable for illustrating vascular proteins of the BBB at the molecular level. This methodology paper is significant since the discovery of BBB therapeutics is highly dependent on visualizing or analyzing the molecular components of BBB.

## Figures and Tables

**Figure 1 ijms-23-07776-f001:**
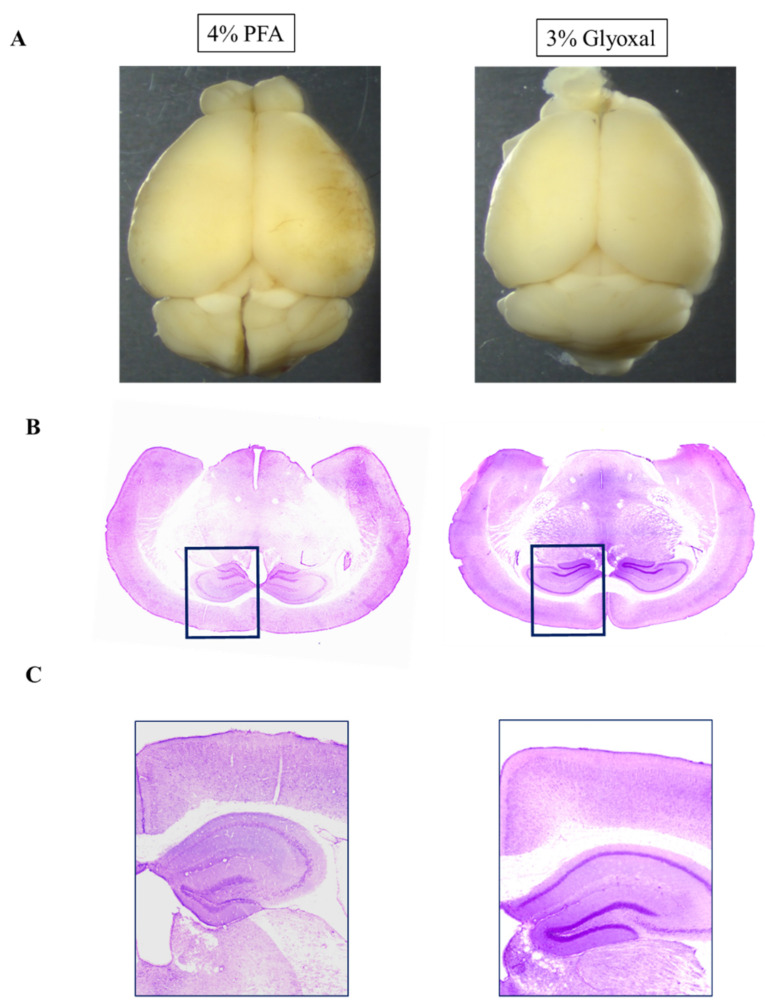
PFA versus Glyoxal fixed mouse brain: Gross Anatomy and H & E staining. (**A**) Comparison of whole mouse brain after 4% PFA and 3% glyoxal fixation (5×). (**B**,**C**) The coronal plane slices from PFA or glyoxal fixed brain sections after Hematoxylin and Eosin staining (5×).

**Figure 2 ijms-23-07776-f002:**
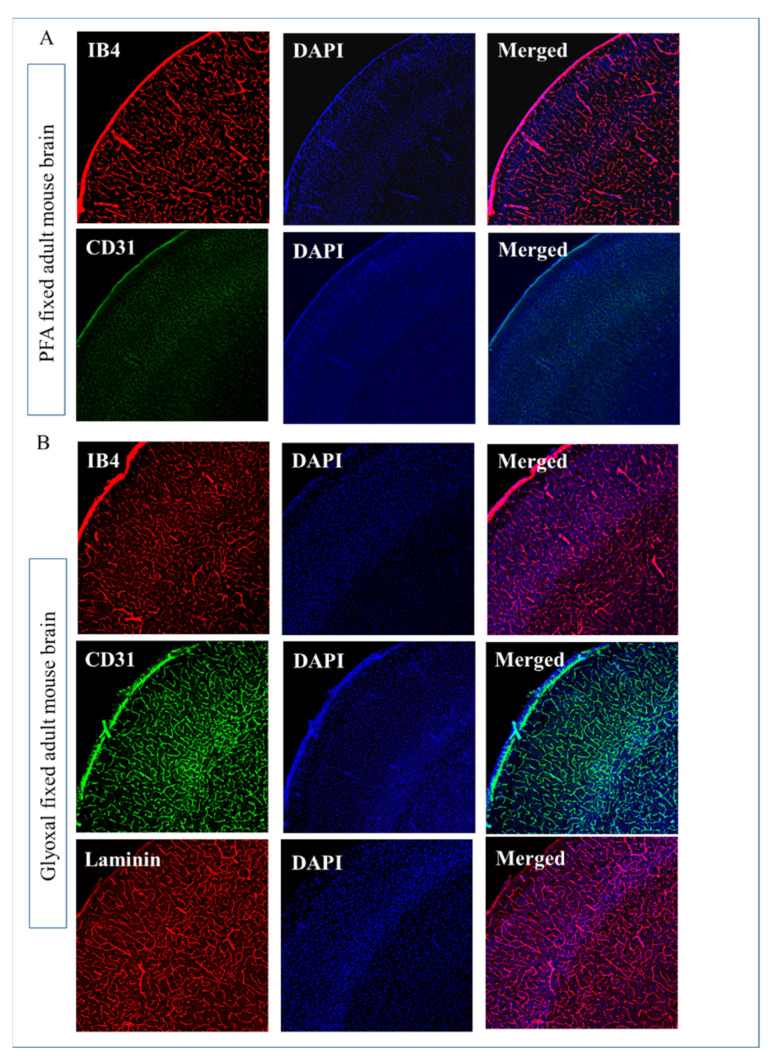
Immunostaining of commonly used antibodies in vascular staining comparison between 4% PFA and 3% glyoxal fixation. (**A**) 35 microns coronal plane slices of PFA fixed brain after IB4 (red) and CD 31(green) staining and nuclei were detected by DAPI nuclear stain. (**B**) The coronal plane slices of glyoxal fixed brain after IB4(red), CD 31 (green), and laminin staining (red), nuclei were detected by DAPI nuclear stain.

**Figure 3 ijms-23-07776-f003:**
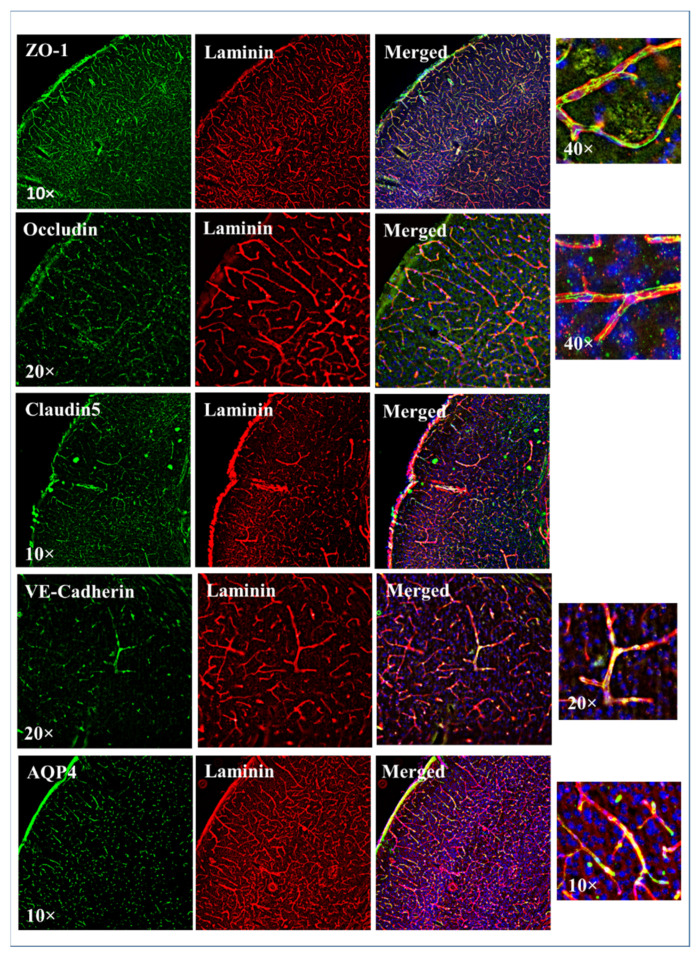
Tight junction and adherens junction staining in 3% glyoxal fixed brain. A total of 35 microns coronal sections from glyoxal fixed brain samples stained with ZO-1, occludin, claudin-5, VE-Cadherin, and aquaporin (AQP4) antibodies as tight junction/BBB markers and laminin or IB4 was used for common vascular staining.

**Figure 4 ijms-23-07776-f004:**
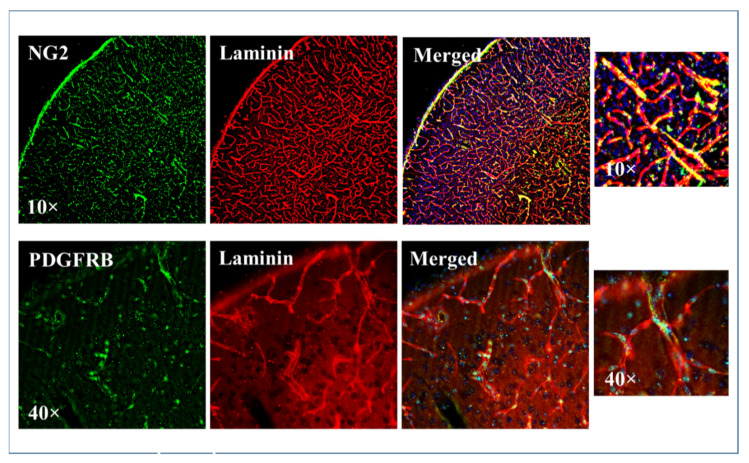
Pericyte staining in 3% glyoxal fixed brain. Pericyte staining of brain coronal plane sections after glyoxal fixation. Pericytes in brain sections were detected by common pericyte markers, NG2 and PDGFRB and laminin was used for vascular staining.

**Figure 5 ijms-23-07776-f005:**
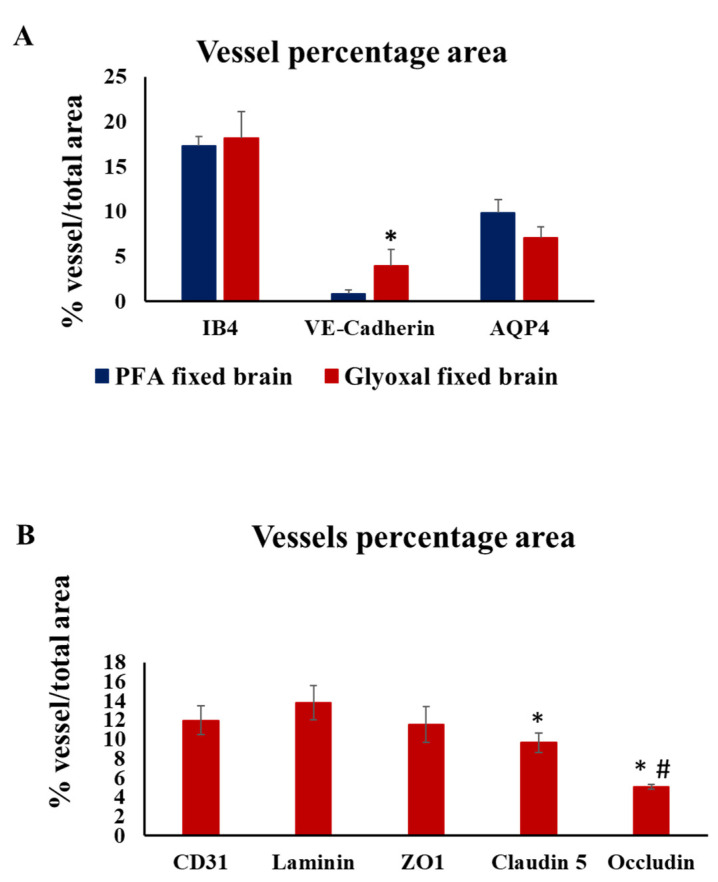
Quantitative analysis of the vascular network. (**A**) Angiotool analysis of percentage vessel area in PFA fixed brain versus glyoxal fixed brain, based on immunofluorescence after staining for vascular markers (IB4-isolectin, VE-cadherin and Aquaporin) and graphical representation of the analysis performed on three different regions of interests. The data shown represent the mean ± SD; statistical analysis was performed with unpaired *t*-test; *n* = 3. * *p* < 0.05 PFA fixed brain sample versus glyoxal fixed brain samples. (**B**) Angiotool analysis of percentage vessel area in glyoxal fixed brain, based on immunofluorescence after staining for vascular markers (CD31, Laminin, ZO-1, Claudin-5, and Occludin) and graphical representation of the analysis performed on three different regions of interest. ANOVA and a post hoc Tukey’s test were performed between groups for finding the statistical significance. A value of *p* < 0.05 was considered statistically significant. * *p* < 0.05 compared to laminin. # *p* < 0.05 compared to CD31.

**Table 1 ijms-23-07776-t001:** Comparison 4% PFA vs. Glyoxal Fixation.

Fixative	4% PFA	Glyoxal
Toxicity	Toxic to handlers [48]	Less toxic [18]
Preparation	Time-consuming	Easy to prepare
Fixation procedure	Intra cardiac perfusion of PFA	Directly putting the tissue in glyoxal
The time needed for fixation	Overnight at 4 °C.	Overnight at RT.
Storage of fixed tissue	4 °C.	4 °C.
Handling floating sections	Easy to handle	A tendency for folding over is high, so needs more care in handling.
Antigen retrieval	Required	Not necessary
Blocking	Required	Not necessary
Vascular Staining quality	Not always satisfactory	High-resolution staining with all of the tested vascular staining’s.
Cost	Less expensive	Moderately Expensive

## Data Availability

Data will be shared upon reasonable request.

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
