# Peer review of "Glyoxal Fixation Is Optimal for Immunostaining of Brain Vessels, Pericytes and Blood-Brain Barrier Proteins"

_ijms, 2022, doi:10.3390/ijms23147776_

Round 1

Reviewer 1 Report

This is an impressive work by the authors, and is certainly a commendable work in identifying an alternative fixative for brain tissue sections especially vascular cells. This may overcome the shortcomings associated with PFA fixation, and facilitate advancement of our knowledge in several health conditions involving the BBB.

I have several suggestions and a few questions for the authors:

1. Why did the authors focus only on vascular staining? It would be useful for the readers to know the answer to this question, and it would be even better to mention if glyoxal-fixation can be useful in other non-vascular staining of brain or other tissues.

2. Reference 9 on line 38 does not support the information it is being cited with. Please recheck.

3. The text in lines 133-137 is out of place in 'Results' and is better suited for the 'Discussion' section.

4. Lines 137-38: please provide complete comparison of incubation times for fixation with PFA or glyoxal.

5. Line 148: CD31 stating is included in Fig. 2A and NOT is Supp. Fig. Please correct.

6. Please rearrange the panels in Supp. figure in the order/pattern they are mentioned in the text.

7. Lines 158-59: Please explain why is Claudin-5 is shown colocalized with IB4, and everything else with laminin in Fig. 3?

8. Line 164: Aquaporin staining for Glyoxal-fixed sections is depicted in Fig. 3, and that for PFA-fixed sections is depicted in Supp. Fig. 1. (Please state correctly.)

9. Line 172: Please show the images from staining for NG2 and PDGFRB for completeness.

10. Lines 185-86: Please rephrase. The sentence is unclear as written. Please add 'in glyoxal-fixed sections' at the end of this sentence for better clarification.

11. Lines 242-43: Would it be more appropriate to replace the marked section with- 'both the fixatives work well for these markers'?

12. Line 252: Please rephrase. Suggestion: In summary, the above evidence supports that despite some limitations, glyoxal is a better alternative to ...

13. There are several language and syntax errors in the manuscript. The authors should consider reviewing the draft for language before resubmission. A pdf file with my annotations for the above comments and other suggestions is attached.

Author Response

Comments

Responses

Reviewer 1

Why did the authors focus only on vascular staining? It would be useful for the readers to know the answer to this question, and it would be even better to mention if glyoxal-fixation can be useful in other non-vascular staining of brain or other tissues.

The authors thank the reviewer for pointing out this.

We have a particular interest in cerebral vasculature as it has distinct anatomy compared to peripheral vessels and is heterogeneous. The structural and molecular analyses of the cerebral vasculature are key to delivering essential information on cerebral circulation, BBB, and injury. The technical limitations in staining vascular components may slow down the progress of cerebrovascular research (Line no.42-26)

Reference 9 on line 38 does not support the information it is being cited with. Please recheck.

The authors thank the reviewer for pointing out this. We have now added appropriate references. Please find new references on Line no.48.

1.      Bogen SA, Vani K, Sompuram SR. Molecular mechanisms of antigen retrieval: antigen retrieval reverses steric interference caused by formalin-induced cross-links. Biotechnic & Histochemistry. 2009 Jan 1;84(5):207-15.

2.      O'Leary TJ, Fowler CB, Evers DL, Mason JT. Protein fixation and antigen retrieval: chemical studies. Biotechnic & Histochemistry. 2009 Jan 1;84(5):217-21.

3.      Hayat, MA. Microscopy, immunohistochemistry, and antigen retrieval methods: Light and electron microscopy. New York, Boston, Dordrecht, London, Moscow: Kluwer Academic/Plenum Publishers; 2001

The text in lines 133-137 is out of place in 'Results' and is better suited for the 'Discussion' section.

The text in lines 133-137 included under the discussion section (Line no. 303-308)

Lines 137-38: please provide complete comparison of incubation times for fixation with PFA or glyoxal.

The time needed for fixation is overnight at 4°C for PFA and overnight at RT for glyoxal. It is mentioned in the Table and included in the text. (Line no. 137-138)

 Line 148: CD31 staining is included in Fig. 2A and NOT is Supp. Fig. Please correct.

Now we have made the necessary correction.

Please rearrange the panels in Supp. figure in the order/pattern they are mentioned in the text.

As per the reviewer's suggestion, we have now rearranged the sup. figures to match the order in the text.

Lines 158-59: Please explain why is Claudin-5 is shown colocalized with IB4, and everything else with laminin in Fig. 3?

We thank reviewer for noting this error. It was a typing error and it is now corrected (as Laminin) in the revised manuscript (Figure. 3)

Line 164: Aquaporin staining for Glyoxal-fixed sections is depicted in Fig. 3, and that for PFA-fixed sections is depicted in Supp. Fig. 1. (Please state correctly.)

The reviewer noted this correctly. We decided to move the PFA fixed-AQP4 staining to supplementary because the PFA fixed brain didn’t show any staining with laminin and thus the figure is incomplete and limited in showing AQP4 colocalization with vascular staining.

Line 172: Please show the images from staining for NG2 and PDGFRB for completeness

NG2 staining image for PFA fixed brain is already in the supplementary figure and now image for PDGFRB also included

Lines 185-86: Please rephrase. The sentence is unclear as written. Please add 'in glyoxal-fixed sections' at the end of this sentence for better clarification.

It is reframed as “Further, we quantified the vessel percentage area for vascular/pericyte proteins including CD31, laminin, ZO-1, claudin5 and occludin in glyoxal fixed brain sections. In glyoxal fixed brain sections, claudin5 showed significantly low vessel percentage area compared to laminin and occludin showed a significant decrease in vessel percentage area compared to laminin and CD31(Figure. 5B)’’. (Line no. 258-262)

Lines 242-43: Would it be more appropriate to replace the marked section with- 'both the fixatives work well for these markers'?

The sentence is corrected as “Finally, the quantification of vascular staining revealed no difference in IB4 and AQP4 between PFA and glyoxal fixed brains indicating both the fixatives work well for these markers.” (Line no. 355-356).

Line 252: Please rephrase. Suggestion: In summary, the above evidence supports that despite some limitations, glyoxal is a better alternative to ...

The sentence is corrected as “In summary, the above evidence supports that despite some limitations, glyoxal is a better alternative to PFA as it is more efficient, specific, easy to fix and highly valuable for illustrating vascular proteins of the BBB at the molecular level”. (Line no. 366-368).

There are several language and syntax errors in the manuscript. The authors should consider reviewing the draft for language before resubmission. A pdf file with my annotations for the above comments and other suggestions is attached.

Corrections are made as suggested

Reviewer 2 Report

The manuscript titled "Glyoxal Fixation Is Optimal for Immunostaining of Brain Vessels, Pericytes, and Blood-Brain Barrier Proteins " by Sithara Thomas et al. aims to compare glyoxal and Paraformaldehyde fixation of brain vessesels. In particular authors compare the different immunostaining of Brain Vessels, Pericytes, and Blood-Brain Barrier Proteins in samples fixed in glyoxal an paraformaldehyde. The glyoxal as a possible substitute for Paraformaldehyde is a rediscovered methods since the use of glyoxal as a substitute for formalin in histological fixations was proposed for the first time in 1943 by  Wicks and  Suntzeff  (Wicks LF, Suntzeff V. GLYOXAL, A NON-IRRITATING ALDEHYDE SUGGESTED AS SUBSTITUTE FOR FORMALIN IN HISTOLOGICAL FIXATIONS. Science. 1943 Aug 27;98(2539):204. doi: 10.1126/science.98.2539.204. PMID: 17843715).

In my opinion the research theme of this manuscript is more appropriate to a journal specialized in methodological analysis.

I do not think this manuscript can be published in IJMS.

Below a few comments

The Representative pictures need to be inserted in the body text.

The section Results and Discussion does not contain the Discussion that is in the section below.

The dilutions of the secondary antibodies are not reported.

The manuscript is poorly edited:

-lack the affiliation for each authors

-different font are present in the manuscript

Author Response

Comments                                                          Response

We understand the reviewer's concern. However, IJMS scope includes breakthrough experimental technical progress of broad interest in Biology and publishing all aspects of molecular research. This is common for interdisciplinary journals such as nature communications.  Also, we would like to point the reviewer's attention to previously published methodology articles in IJMS.

1.      Lin GM, Lai YH, Audira G, Hsiao CD. A simple method to decode the complete 18-5.8-28S rRNA repeated units of green algae by genome skimming. International journal of molecular sciences. 2017 Nov 6;18(11):2341.

2.      Herrmann J, Babic M, Tölle M, Eckardt KU, van der Giet M, Schuchardt M. A novel protocol for detection of senescence and calcification markers by fluorescence microscopy. International journal of molecular sciences. 2020 May 14;21(10):3475.

3.      Lo Piccolo L, Bonaccorso R, Onorati MC. Nuclear and cytoplasmic soluble proteins extraction from a small quantity of Drosophila’s whole larvae and tissues. International Journal of Molecular Sciences. 2015 Jun 1;16(6):12360-7.

4.      Nardin, Chiara, Abraham Tettey-Matey, Viola Donati, Daniela Marazziti, Chiara Di Pietro, Chiara Peres, Marcello Raspa, Francesco Zonta, Guang Yang, Maryna Gorelik, Serena Singh, Lia Cardarelli, Sachdev S. Sidhu, and Fabio Mammano. 2022. “A Quantitative Assay for Ca2+ Uptake through Normal and Pathological Hemichannels.” International Journal of Molecular Sciences 2022, Vol. 23, Page 7337 23(13):7337.

The Representative pictures need to be inserted in the body text.

As suggested by the reviewer, now we have inserted the representative pictures in the body text

The section Results and Discussion does not contain the Discussion that is in the section below.

The section Results and Discussion is removed and are included as separate sections in the manuscript. (Line no. 134-300 and Line no. 302-364)

The dilutions of the secondary antibodies are not reported.

It was 1:200 and the information has been added to the manuscript (Line no. 121)

The manuscript is poorly edited:

    -lack the affiliation for each author

    -different font are present in the manuscript

Author affiliations are inserted correctly (Line no. 3-4)

Manuscript is corrected to have only one font

Reviewer 3 Report

The manuscript entitled "Glyoxal Fixation Is Optimal for Immunostaining of Brain Vessels, Pericytes, and Blood-Brain Barrier Proteins" describes improvements in the staining protocol of brain tissues and cells. The authors have proven that the minor changes in the reagent used for fixation may be beneficial for the effectiveness of staining and the quality of the obtained images. Moreover, the proposed reagent is less toxic than PFA. The innovativeness of the work is moderate, however, the manuscript is written in the correct form and the methods are provided with details. I support the manuscript for its publication in IJMS after some minor revisions such as correction of typos, etc.

Author Response

Reviewer 3

I support the manuscript for its publication in IJMS after some minor revisions such as correction of typos

Typographical errors have been corrected

Round 2

Reviewer 2 Report

The revised version of the manuscript can be accepted